# Construction of Safety-Management Platform for Chemical Enterprises Based on the Immune System Model

Xiongjun Yuan [1,2], Bingjie Wang [1], Xin Guan [1], Yuqin Wang [1], Othmane Chemsi [1], Jiaxuan Zhang [1] and Xiaoyu Chen [1,*]

[1] School of Safety Engineering, China University of Mining and Technology, Xuzhou 221116, China; yxj@cczu.edu.cn (X.Y.); TS21120142P31@cumt.edu.cn (B.W.); guan@cumt.edu.cn (X.G.); TS19120032A31TM@cumt.edu.cn (Y.W.); othmanechemsicumt@gmail.com (O.C.); 16194878@cumt.edu.cn (J.Z.)

[2] School of Environmental and Safety Engineering, Changzhou University, Changzhou 213164, China

[*] Correspondence: cxy@cumt.edu.cn

**Abstract:** Identifying risk factors and improving safety "immunity" is beneficial to chemical enterprises to reduce the occurrence of production accidents. In this paper, an immune system model has been built according to the bionics principle, and the corresponding antigen–antibody index system based on hidden dangers and treatment measures has been established, including 8 factor layers and 33 index layers. Then, the immune evaluation model of production safety was built using index weight, and an evaluation model of immune response ability for production safety was constructed based on grey system theory. Finally, aiming to identify immune deficiency through numerical analysis, a safety-management platform focused on people management, visual monitoring, and danger warning was constructed. The results verify the suitability of the immune correlation principle applied in chemical enterprises, which makes an instructive contribution to the construction of information management platforms for chemical enterprise safety production.

**Keywords:** chemical enterprise; accident risk; immune system model; safety-management platform; grey system theory



## 1. Introduction

With the development of the world economy, chemical industry has become an important pillar industry of national economy. Although the chemical industry brings huge economic benefits, due to complex production process and various raw materials and waste products, serious accidents often occur. Statistically, from 2006 to 2015, there were 125 production safety accidents in chemical enterprises in China, resulting in 542 deaths, 438 injuries, and 21 missing persons, with an average of 12.5 chemical production accidents per year, and an average of 54.2 deaths per year due to chemical production accidents [1,2]. These data reveal deficiencies in the safety management and monitoring systems of the industry, as well as the lack of safety awareness and professional abilities among employees [3,4]. At the same time, the expansion of the chemical enterprises makes it impossible to correctly predict the trend of accidents, which can result in uncontrolled risks and hidden dangers.

Since the 1970s, researchers have made a lot of progress in immune system research. Farme et al. firstly introduced the concept of immunity and opened the prelude to the application of immune systems in the engineering field [5]. Varela put forward the viewpoint that immune systems adapted to different situations by producing different antibodies and mutations, laying a foundation for solving engineering problems by applying immune mechanisms [6]. Dasgupta et al. showed unprecedented research on artificial immune systems in the field of information intelligence [7]. Shi et al. [8] made a comprehensive comparison and analysis between immune systems and emergency management, and proposed

an emergency management system based on immune recognition, response, feedback, and memory. At present, immune systems are widely used in many fields, such as fault diagnosis [9], pattern recognition [10–12], dynamic monitoring, and early warning [13–15].

The application of immune systems in the safety production management of chemical enterprises is still in its infancy [16]. In the complex dynamic space of personnel, equipment, environment, and management of chemical enterprises, it is essential to apply the immune system to promote enterprise immune defense, as well as to build a system of accident prevention and early warning. In this paper, by referring to the mechanism of the biological immune system, the immune mechanism is integrated into the production safety management of chemical enterprises to form the immune correlation principle of accident risk and establish an antigen–antibody model of hidden dangers. On this basis, the existing problems and unknown dangers in the safety management of chemical enterprises have been analyzed. Moreover, the immune system and evaluation model of chemical enterprises safety production are established. The evaluation model of immune response ability is designed based on grey system theory [17], and the maturity and immune response ability are quantitatively analyzed. This is conducive to identifying immune deficiency and clarifying the improvement direction of safety management and accident-prevention approaches in chemical enterprise production, which has important significance for improving the safety management layer of the whole chemical industry.

## 2. The Correlation Principle of Accident Risk Immunity in Chemical Enterprises

The biological immune system is a specific defense system against pathogen invasion, which is a multilayer, adaptive system composed of immune-active molecules, immune cells, immune tissues, and organs. Once organisms are attacked by pathogens, the immune system quickly produces an immune response, identifying and killing pathogens, then it forms immune memory and generates immune feedback.

According to the immunological perspective [18], the similarities in structure, object, function, and mechanism between the safety-management system of chemical enterprises and the biological immune system were compared (Figure 1). Specifically, for organization structure, the immune system is composed of immune cells, molecules, and organs, while the production management system is composed of human, machine, environment, and management. Both are sophisticated systems. For the target object, antigens in the body and the environment are the objects of the immune system, while the hazard sources inside and outside the chemical enterprise are the objects of the production management system. From the aspect of function, the basic function of the immune system is to eliminate harmful antigens and avoid diseases, while the basic function of the production management system is to eliminate the hidden danger of accidents and avoid accidents, so they have similar functions. Finally, these two systems have similar mechanisms of adaptive and constant change; the immune system adjusts to changes in the body and the body adjusts to the immune system, while the production management system adjusts and improves with the change of enterprise production, promoting the improvement of other systems in chemical enterprises. Consequently, the immune mechanism can be used to optimize the chemical enterprise safety production management model.

Similar to the work mechanism of the biological immune system [19,20], the early-warning system of chemical enterprises solves safety problems by reflecting a continuous process of "learning", "memory", and "feedback" [21]. Based on the actual functions, the safety-management system of chemical enterprises can be divided into three parts, as shown in Figure 2: monitoring and recognition of production accident antigen; early warning and feedback of inducing factor; and antibody immunity.

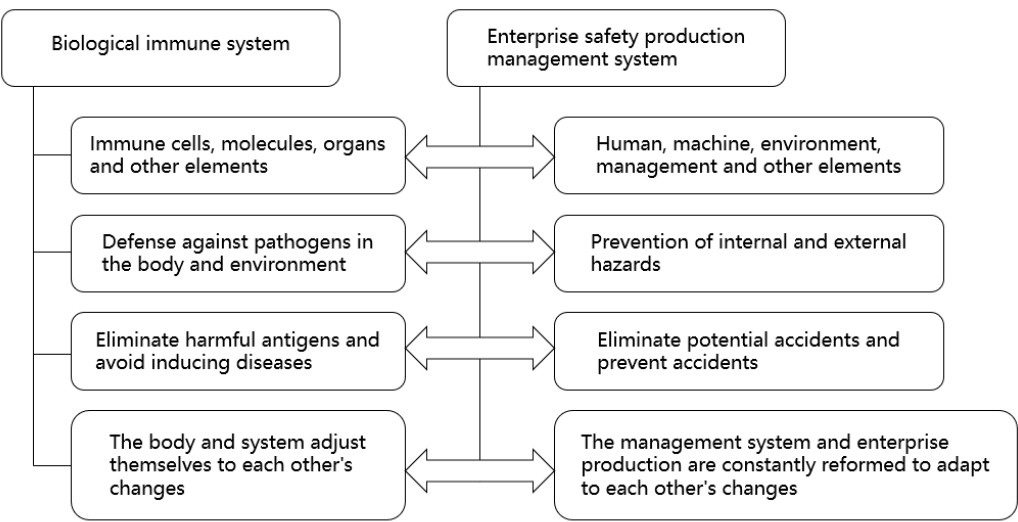

**Figure 1.** Correlation between immune system and enterprise security management system.

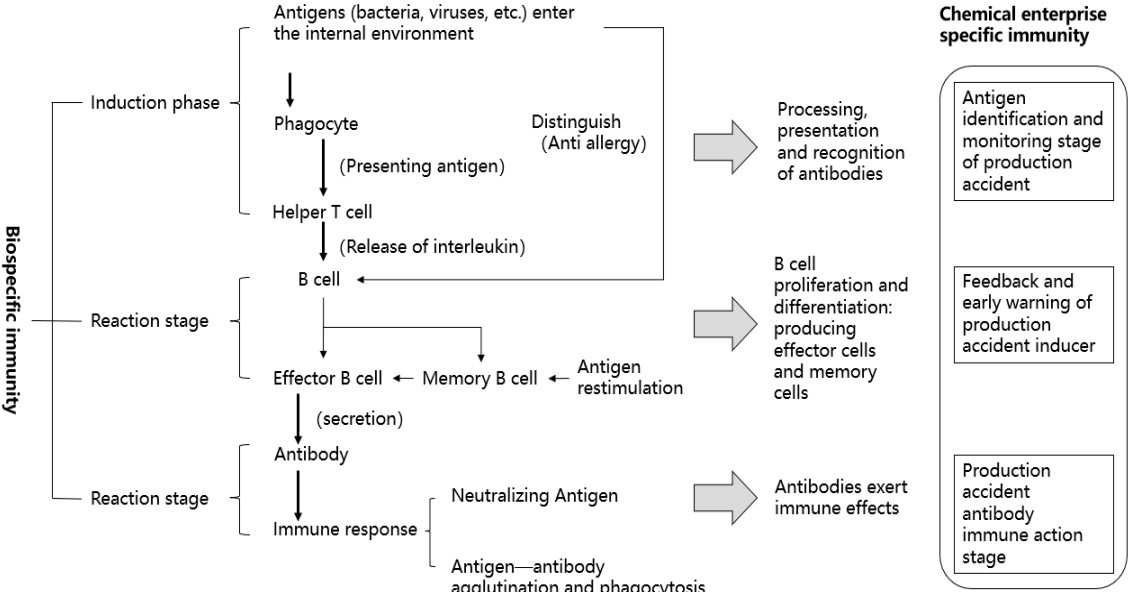

**Figure 2.** Similarity of enterprise-specific immunity and immune mechanism.

From the similarity analysis of system structure and prevention mechanism, it is obvious that the safety-management system of chemical enterprises is closely related to biological immune system, and its function is to eliminate risk factors and prevent the occurrence of "pathological changes" in enterprises. The construction of the immune system model of production safety in chemical enterprises can be divided into three parts: a danger-treatment model based on the antigen–antibody model; the immune evaluation model of safe production; and the immune response ability evaluation model. This model can effectively identify the risks of "external antigens" and "internal lesions", identify immune deficiencies or vulnerabilities, and put forward the corresponding control measures and early-warning scheme.

## 3. Antigen–Antibody Correspondence Model of Immune System in Chemical Enterprise Production

Dangerous and damaging factors are common aspects in current chemical industry processes [22]. Production crises are usually caused by accident-inducing factors attacking the enterprise's normal operation and trying to control some inherent behaviors, safety

equipment, components, and safety organization. Enterprise production crisis is the result of hazard factors interfering with normal operations and an attempt to control some inherent behavior, equipment, or components. In this case, the internal "defense team" of the production system will warn the attacked parts in time, inform the management personnel to take technical measures as soon as possible, avoid the control of accident infection, prevent the intrusion of accident-inducing factors, and ensure the smooth progress of production. Therefore, according to the existing accident cause statistics and relevant safety regulations, combined with the risk identification principle of biological immune system [23], a four-layer antigen–antibody-corresponding model of dangers–measures, comprising four major factors—human, machine, environment, and management—was established, as shown in Table 1.

**Table 1.** Safety production index system of chemical enterprises based on immune model.

| Target Layer | Criterion Layer | Element Layer | Index Layer |
|---|---|---|---|
| **Safe production in chemical enterprises** | **Antigen G** | Unsafe behavior G1 | Staffing G11 |
| | | | Illegal construction G12 |
| | | | Personal protection G13 |
| | | | Personal quality G14 |
| | | Unsafe state G2 | Object failure G21 |
| | | | Disorderly traffic G22 |
| | | | Equipment mechanical damage G23 |
| | | | Violation of distance and placement G24 |
| | | | Lack of protection G25 |
| | | Adverse operating environment G3 | Major hazard monitoring G31 |
| | | | Risk warning value exceeds G32 |
| | | | Environmental pollution G33 |
| | | | Occupational health hazards G34 |
| | | Lack of Safety Management G4 | Safety education G41 |
| | | | Security screening governance G42 |
| | | | Work safety supervision G43 |
| | | | Safety identification and Emergency G44 |
| | **Antibody B** | Safe behaviors B1 | Personnel on-job management B11 |
| | | | Behavior safety observation B12 |
| | | | Personnel safety training B13 |
| | | | Risk control of hazardous operations B14 |
| | | Safe state B2 | Monitoring and early warning of major hazard sources B21 |
| | | | Monitoring and warning of high-risk processes B22 |
| | | | Video surveillance of vital parts B23 |
| | | | Inherent facility risk management B24 |
| | | Good working environment B3 | Risk monitoring map B31 |
| | | | Monitoring and warning of toxics B32 |
| | | | Occupational disease risk monitoring B33 |
| | | Safety management B4 | Risk prevention and control objective management B41 |
| | | | Risk identification and control B42 |
| | | | Risk investigation and management B43 |
| | | | Emergency management B44 |
| | | | Training learning assessment B45 |

The first layer is the target layer that leads to the achievement of the overall goal of enterprise production safety, that is, safe production. The second layer is the criterion layer,

that ensures the realization of the overall goal, which is composed of two subsystems, antigen (G) and antibody (B). The third layer is the element layer, which consists of four basic immune objects, namely human behavior, state of things, environmental conditions, and safety management. In this, G and B have four secondary layers of elements each, so there are eight elements. The fourth layer is the index layer (basic layer), which is the most basic level of the whole system, including all specific indexes of early warning which are directly measurable factors of diagnosis.

Based on the bio-immune mechanism, with the prevention of production safety accidents in chemical enterprises as the goal, the overall analysis of the production structure of the enterprise was firstly conducted from the perspective of bionics. Then, combining the similarity between the biological immune system theory and the safety production system of chemical enterprises, we constructed a hidden danger identification and early-warning and risk-assessment model for the safety immune system of chemical enterprises. On the basis of the safety information database of chemical enterprises, the results of enterprise hazard source research, chemical production safety regulations, and safety expert risk assessment were integrated, and then the antigen–antibody indicators (G11–G44 and B11–B45, respectively) of the immune system of production accidents in chemical enterprises were extracted. Finally, hierarchical analysis and expert scoring were applied to determine the weights of indicators set by the evaluation index system to achieve the maturity assessment of production safety immunity of chemical enterprise.

## 4. Construction Principle of Immune System Model of Safe Production in Chemical Enterprise

### 4.1. Determination of Index Weight of Immune Model in Safe Production

Since the difference of evaluation index will affect the final evaluation result, an expert scoring method and analytic hierarchy process were comprehensively used to determine the weight of elements at each layer of the index system [24]. The index scoring range was divided into five grades: "very poor", "poor", "general", "good", and "very good", with corresponding score values of 1, 2, 3, 4, and 5, respectively. A 1–5 scale method was adopted to score the mean value of each item, the second and first layer index judgment matrix were constructed according to $G_{ij} = (m_{1ij} + m_{2ij} + \ldots + m_{10ij})/10$ and $G_i = \sum W_{ij} \cdot G_{ij}$.

Where $i = 1, 2, 3, 4$; $j = 1, 2, 3, \ldots, n$; $G_{ij}$ is the mean score of the secondary index, $m_1$–$m_{10}$ are scores by experts, and $W_{ij}$ is the weight of index layer (second layer).

### 4.2. Evaluation Principle of Immune Model in Safe Production

a.    Calculation of immune indexes' contributions

Principal component analysis (PCA) mainly studies the multiple factors concentration and main information extraction, aiming to reveal most of the information from original indexes by a few key immune indexes. The calculation steps are as follows:

In the first step, according to the existing four immune objects and $n$ evaluation indexes, the evaluation results of each immune index were normalized.

$$S_k^2 = \frac{1}{n-1} \sum_{k=1}^{n} (X_{ik} - \overline{X_k})^2 \tag{1}$$

where $i = 1, 2, 3, 4$; $k = 1, 2, 3, \ldots, n$; $X_{ik}$ is value of certain index; $\overline{X_k}$, $S_k^2$ are mean value and standard deviation of evaluation indexes, respectively.

The second step is to construct the immune index matrix $X$:

$$X = (X_{ik})_{4 \times n} = \begin{bmatrix} X_{11} & \ldots & X_{1n} \\ \vdots & \ddots & \vdots \\ X_{41} & \ldots & X_{4n} \end{bmatrix} \tag{2}$$

The third step is the orthogonalization of immune index matrix. Here, $L_{ik}{}^T$ is a set of unit vectors ($L_{ik}{}^T = [L_{i1}, L_{i2}, \ldots, L_{in}]^T$ and $\sum_{k=1}^{n} L_{ik}{}^2 = 1$), through which new comprehensive variables $R_i$ ($i = 1, 2 \ldots 8$) are constructed:

$$\begin{cases} R_1 = L_{11}X_1 + L_{12}X_2 + \ldots L_{1n}X_n \\ R_2 = L_{21}X_1 + L_{22}X_2 + \ldots L_{2n}X_n \\ \qquad \vdots \quad \vdots \quad \vdots \\ R_4 = L_{41}X_1 + L_{42}X_2 + \ldots L_{4n}X_n \end{cases} \tag{3}$$

where $R_1$, $R_2$ ... $R_4$ represent the immune index after normalization.

The fourth step is to calculate the eigenvalue $\lambda$ and the eigenvector $\alpha$ of vector group $R_i$, and the eigenvalue $\lambda_i$ is the variance of the principal component $R$. The principal component $\lambda_i > 1$ is usually selected for analysis.

Fifth, the principal component index is determined according to the correlation coefficient $\beta_i = \sqrt{\lambda_i}\alpha_{ik}$ between the principal immune index $R$ and variable $X_i$. In order to ensure that $R$ has most of the original information, the cumulative contribution $\theta_i \geq 90\%$.

$$\theta_i = \sum_{k=1}^{n} \beta_i{}^2 = \sum_{k=1}^{n} \lambda_k \alpha_k{}^2 \tag{4}$$

where $\beta_i$ is the immune index load, $\alpha_{ik}$ is the $i$th value in the $k$th eigenvector, and $\theta_i$ is the contribution of the $i$th immune index to the immune system.

b.　Layer analysis and evaluation of immune indexes

In order to achieve safe production immune maturity evaluation, a coordinate system is established according to the contribution of immune factors (B) and antigenic factors (G), and the energy level diagram of accident-inducing immune factors is thus drawn in Figure 3. The center of the circle is the core of the immune system, and the outside of the circle is the external environment. According to the contribution of the immune factor to the immune system (the combined antigen–antibody score value), it is divided into five levels: initial level I [0.0, 0.2], defined level II [0.2, 0.4], repetition level III [0.4, 0.6], continuation level IV [0.6, 0.8], and optimization level V [0.8, 1.0]. For the immunity factor (B), the closest internal level V has the highest energy level, indicating the better safety status of the chemical company. The outermost energy level I is the lowest, which is also the highest hazard level. The further away the accident immune factor is from the control target (center of the circle), the lower the degree of control of the antigen by that antibody, i.e., the lower the energy level. For the antigenic factor (G), which induces harm, the energy level diagram is interpreted in the opposite way. Thus, the degree of control of the incident immune system over the harmful factor (antigen G) depends on the positional state of the corresponding immune factor (antibody B), with each antibody being independent, interrelated, and mutually constrained.

When the accident immune system is in an unstable state, the immune indexes will be "removed" or "missing". On the one hand, these manifest in the problems of people, things, environment, and management, which tend to cause accidents in the production process [25]. On the other hand, from the perspective of the accident-induced energy layer (G), when the enterprise's immunity ability is weak and the maturity of safety production immunity is low, the harmful factors will "gather" or "concentrate", while the immune indexes will be excluded, and hence the hidden danger of safety production accidents will occur.

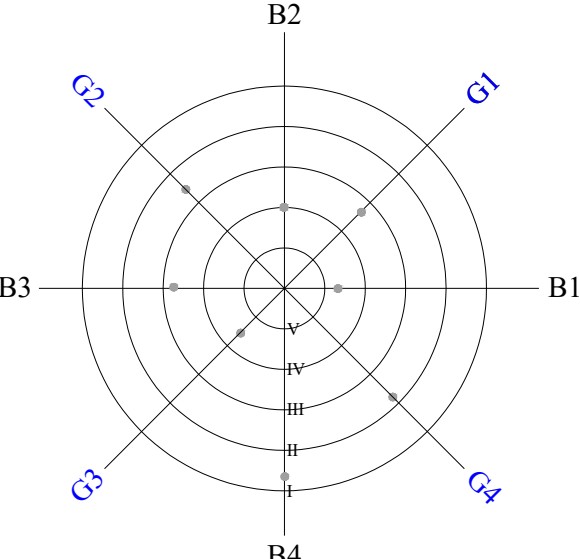

**Figure 3.** Energy layer diagram of accident-inducing immunity factor.

c.  Theory of production safety immune response ability evaluation

Grey system theory refers to the quantitative analysis process of whitening of grey numbers [26], mainly analyzing the effect of immune system on antigen in safe production of chemical enterprises. The specific assessment steps are as follows:

First, each evaluation index of antibody criteria layer is quantitative processed, n risk evaluation indexes are scored by m experts according to the scoring method. The grading of evaluation is divided into "very good", "better", "general", "poor", and "very poor", corresponding to risk layer $e$ = 1, 2, 3, 4, 5, respectively, according to the interval median [10,8], [8,6], [6,4], [4,2], [2,0], rated on a 10-point scale. In order to quantify risk, the model of expert evaluation is usually adopted, namely gray evaluation theory to relevant experts, where the evaluation object $i \in \{1, 2, \ldots, n\}$. For a specific project, the risk item is generally chosen as the evaluation index, and the evaluation index is denoted, where $j \in \{1, 2, \ldots, m\}$. So, the sample matrix $A = Aij$, $i \in \{1, 2, \ldots, n\}$, $j \in \{1, 2, \ldots, m\}$ is built.

$$A = \begin{Bmatrix} A_{11} & A_{12} & \ldots & A_{1m} \\ A_{21} & A_{22} & \ldots & A_{2m} \\ \ldots & \ldots & \ldots & \ldots \\ A_{n1} & A_{n2} & \ldots & A_{nm} \end{Bmatrix} \tag{5}$$

The second step is to establish the whitening weight function and calculate the grey evaluation coefficient $C^e$ in order to accurately quantify the grade and score of the evaluation index: $C^e = C^{e1} + C^{e2} + C^{e3} + C^{e4} + C^{e5}$ (Table 2).

The third step is to normalize the grey evaluation coefficient $Q^{em} = C^{em}/C^e$ ($m$ = 1, 2, 3, 4, 5), and constitute the grey evaluation weight vector $Q^e = [Q^{e1}, Q^{e2}, Q^{e3}, Q^{e4}, Q^{e5}]$, then the grey evaluation matrix of all indexes at each layer is constructed [27]:

$$Q = \begin{bmatrix} Q_1{}^{e1} & Q_1{}^{e2} & Q_1{}^{e3} & Q_1{}^{e4} & Q_1{}^{e5} \\ Q_2{}^{e1} & Q_2{}^{e2} & Q_2{}^{e3} & Q_2{}^{e4} & Q_2{}^{e5} \\ \ldots & \ldots & \ldots & \ldots & \ldots \\ Q_n{}^{e1} & Q_n{}^{e2} & Q_n{}^{e3} & Q_n{}^{e4} & Q_n{}^{e5} \\ \ldots & \ldots & \ldots & \ldots & \ldots \\ Q_m{}^{e1} & Q_m{}^{e2} & Q_m{}^{e3} & Q_m{}^{e4} & Q_m{}^{e5} \end{bmatrix} \tag{6}$$

**Table 2.** Whitening weight functions.

| Gray Scale | Category | Grey Number | White Function | Functional Diagram | Corresponding Evaluation Coefficient |
|---|---|---|---|---|---|
| $e = 1$ | Very good | $\otimes \in [8, 10)$ | $f_1(x) = \begin{cases} 0, x \notin [2, +\infty) \\ \frac{x-2}{8}, x \in [2, 10) \\ 1, x \in [100, +\infty) \end{cases}$ |  | $e = 1, C^{e1} = \sum\limits_{i=1}^{m} f_1(a_{in})$ |
| $e = 2$ | Good | $\otimes \in [6, 8)$ | $f_2(x) = \begin{cases} 0, x \notin [2, 10) \\ \frac{x-2}{6}, x \in [2, 8) \\ \frac{10-x}{2}, x \in [8, 10) \end{cases}$ |  | $e = 2, C^{e2} = \sum\limits_{i=1}^{m} f_2(a_{in})$ |
| $e = 3$ | Ordinary | $\otimes \in [4, 6)$ | $f_3(x) = \begin{cases} 0, x \notin [2, 10) \\ \frac{x-2}{4}, x \in [2, 6) \\ \frac{10-x}{4}, x \in [6, 10) \end{cases}$ |  | $e = 3, C^{e3} = \sum\limits_{i=1}^{m} f_3(a_{in})$ |
| $e = 4$ | Low | $\otimes \in [2, 4)$ | $f_4(x) = \begin{cases} 0, x \notin [1, 8) \\ x - 1, x \in [1, 2) \\ 1, x \in [2, 4) \\ \frac{8-x}{4}, x \in [4, 8) \end{cases}$ |  | $e = 4, C^{e4} = \sum\limits_{i=1}^{m} f_4(a_{in})$ |
| $e = 5$ | Very low | $\otimes \in [0, 2)$ | $f_3(x) = \begin{cases} 1, x \in [0, 1) \\ 2 - x, x \in [1, 2) \\ 0, x \notin [0, 2) \end{cases}$ |  | $e = 5, C^{e5} = \sum\limits_{i=1}^{m} f_5(a_{in})$ |

The fourth step is to calculate the evaluation result of immune response ability of evaluation index according to the score of rating $R^e = [Q^{e1}, Q^{e2}, Q^{e3}, Q^{e4}, Q^{e5}][9,7,5,3,1]^T$.

The fifth step is to set the weight matrix of indexes at each layer as $w = [w_1, w_2, w_3, \ldots, w_n]$, and the evaluation results of overall immune response ability of factor and index layer are calculated as $R = w = [w_1, w_2, w_3, \ldots, w_n][R_1^e, R_2^e, R_3^e, R_4^e, R_5^e]^T$.

## 5. Construction of Immune System Model for Safe Production in Chemical Enterprises

### 5.1. Weight Calculation of Safety Production Immune Model Index

a.  Weight calculation of immune model index

Because different evaluation indexes have different influences on the final evaluation results, this paper determines the weight of the evaluation indexes by using an expert scoring method and the analytic hierarchy process (AHP). Take the weight of secondary immunity index as an example. When AHP is used to calculate the weight of antibody index, firstly, the judgment matrix of B11–B14 needs to be constructed (constructed by MATLAB), and the judgment matrix is obtained by pairwise comparison of elements (Table 3). The judgment matrix is constructed by first calculating the average value of each analysis item, and then dividing it by the average value to obtain the judgment matrix. It is obvious that a higher average means a higher importance and a higher weight.

**Table 3.** AHP layer analysis judgment matrix.

| Average | Item | B11 | B12 | B13 | B14 |
|---|---|---|---|---|---|
| 2.200 | B11 | 1 | 0.957 | 0.759 | 0.611 |
| 2.300 | B12 | 1.045 | 1 | 0.793 | 0.639 |
| 2.900 | B13 | 1.318 | 1.261 | 1 | 0.806 |
| 3.600 | B14 | 1.636 | 1.565 | 1.241 | 1 |

Secondly, the characteristic roots and weights are calculated. Eigenvector value, weight value and maximum eigenvector are calculated, and the maximum feature root value *CI* is obtained, which is used for the following consistency test of the maximum characteristic root. In order to describe the weight of each index one by one, the sum product method combined with the square root method are used here to study AHP.

As shown in Table 4, the fourth order judgment matrix is constructed for 4 indexes of B11–B14 and the AHP method (sum product method) is studied. The feature vectors obtained are (0.800, 0.836, 1.055, 1.309), and the corresponding weight values $W_{Bij}$ are 20.000%, 20.909%, 26.364%, and 32.727%, respectively. In addition, the maximum eigenroot can be calculated as 4 by combining the eigenvectors. For immune risk prevention and control, the importance of safety prevention of factors B1 regulating unsafe behaviors of people is ranked from large to small as B14 > B13 > B12 > B11. The expert scoring and AHP weight analysis show that chemical enterprises should focus on strengthening the risk control of dangerous operations in the production process and the safety training of employees in the area for the management of unsafe behavior. The contribution of the two factors to the immune maturity of unsafe behavior accidents is close to 60%.

**Table 4.** AHP layer analysis results.

| Item | Eigenvector | Weighted Value | Maximum Eigenvalue | CI Value |
|---|---|---|---|---|
| B11 | 0.800 | 20.000% | | |
| B12 | 0.836 | 20.909% | 4.000 | 0.000 |
| B13 | 1.055 | 26.364% | | |
| B14 | 1.309 | 32.727% | | |

It is noticeable when the judgment matrix $n \geq 3$, expert scoring is prone to subjective bias and can be judged by consistency test [28]. The random consistency ratio (*CR = CI/RI*) was used as a measure, *CI* is maximum characteristic root and *RI* is random consistency, where *CI* = ($\lambda_{max} - n$)/($n - 1$) and *RI* calculation results are listed in Table 5. If the *CR* value is smaller (generally *CR* < 0.1), then it indicates that the consistency of the judgment matrix is better, and the consistency test of the judgment matrix can be passed. However, if *CR* value > 0.1, it indicates that consistency test fails, and AHP analysis should be carried out after proper adjustment of judgment matrix. In this study, a fourth-order judgment matrix is constructed, and the RI value of random consistency can be queried in the table above as 0.890, so *CR* = 0 < 0.1, the judgment matrix meets the consistency test.

**Table 5.** Random consistency *RI* table.

| *n* | 3 | 4 | 5 | 6 | 7 | 8 | 9 | 10 | 11 | 12 | 13 | 14 | 15 | 16 |
|---|---|---|---|---|---|---|---|---|---|---|---|---|---|---|
| *RI* | 0.52 | 0.89 | 1.12 | 1.26 | 1.36 | 1.41 | 1.46 | 1.49 | 1.52 | 1.54 | 1.56 | 1.58 | 1.59 | 1.59 |

| *n* | 17 | 18 | 19 | 20 | 21 | 22 | 23 | 24 | 25 | 26 | 27 | 28 | 29 | 30 |
|---|---|---|---|---|---|---|---|---|---|---|---|---|---|---|
| *RI* | 1.606 | 1.613 | 1.621 | 1.629 | 1.636 | 1.640 | 1.646 | 1.650 | 1.656 | 1.659 | 1.663 | 1.667 | 1.669 | 1.672 |

Based on the calculated index layer weight $W_{ij}$ (the second layer), calculate the factor layer weight $W_i$ (the first layer). Take the average $G_{ij}$ of each second-layer index score, the input data of index layer is the weighted average of average score of index elements and

corresponding weights $G_i = \sum W_{ij} \cdot G_{ij}$. The weight calculation of the element layer or the criterion layer is similar. The factor layer needs to input judgment matrix separately to calculate the weight. The factor layer and index layer are individually calculated to obtain the weight, and the final weight value of each index to the target layer can be obtained by multiplying them.

Taking the calculation of $W_{Gi}$ as an example, using AHP to calculate weights, the first step is to create a judgment matrix, calculate the weighted average value of all indexes, and then divide the average value to obtain the judgment matrix (Table 6).

**Table 6.** AHP layer analysis judgment matrix.

| $G_i$ Weighted Average | Item | G1 | G2 | G3 | G4 |
|---|---|---|---|---|---|
| 2.940 | G1 | 1 | 1.059 | 0.981 | 1.085 |
| 2.777 | G2 | 0.944 | 1 | 0.926 | 1.025 |
| 2.997 | G3 | 1.020 | 1.079 | 1 | 1.106 |
| 2.709 | G4 | 0.922 | 0.976 | 0.904 | 1 |

Similarly, the weights of antibody factor $W_{Bi}$ = (0.24432, 0.28723, 0.23232, 0.23613); $W_{B1j}$ = (0.0489, 0.0511, 0.0644, 0.0800); $W_{B2j}$ = (0.0868, 0.0779, 0.0757, 0.0468); $W_{B3j}$ = (0.0655, 0.1013, 0.0655); $W_{B4j}$ = (0.0399, 0.0538, 0.0556, 0.0434, 0.0434). The weights of antigen $W_{Gi}$ = (2.939993, 2.776811, 2.997391, 2.70926); $W_{G1j}$ = (0.03977, 0.08657, 0.06083, 0.07019); $W_{G2j}$ = (0.04404, 0.04756, 0.04580, 0.05108, 0.05461); $W_{G3j}$ = (0.07986, 0.07758, 0.05704, 0.04792); $W_{G4j}$ = (0.05490, 0.06368, 0.06149, 0.05710).

According to the above calculation, the weight of the early-warning index system of chemical enterprises safety production based on antigen G and antibody B was obtained. The specific calculation results are shown in Table 7.

*5.2. Construction of Immune Evaluation Model for Safety Production*

a.  Contribution of immune indexes

The five principal component factors are screened according to the criterion $\lambda > 1$ and the variance explanation rate is 34.665%, 18.256%, 16.062%, 11.034%, and 7.967%, respectively. The cumulative variance explanation rate is as high as 87.983% < 90%, so the factor needs to be adjusted. Four new comprehensive variables $\{R_1, R_2, R_3, R_4\}$ are finally extracted as representative variables of immune index evaluation according to the auxiliary judgment of the scree plot (as shown in Figure 4).

The variances of $R_1$, $R_2$, $R_3$, and $R_4$ are 5.546, 2.921, 2.570, and 1.765, respectively. $R_1$ is the "main direction", reflecting the most obvious characteristics of all the 16 antibody factors, which maximizes the information contained in the immune system of the production accident. It also avoids the limitation that the immune system of production accidents is difficult to determine, which is caused by the correlation of various immune antibody factors.

The comprehensive score values of antibody index layer B1–B4, namely the contribution degree $\theta$ of immune function to safety production accidents, are 2.33762, 2.61145, 1.53572, and 3.23159, respectively, and the final score of immune maturity in safe production of chemical enterprises is 0.6114.

Similarly, four cofactors are extracted from 17 antigen factors, and the variance explanation rate is 28.799%, 17.074%, 15.057%, and 12.129%, respectively. The cumulative variance explanation rate is 73.059% < 90% (Figure 5).

The variances of $R_1$, $R_2$, $R_3$, and $R_4$ are 4.896, 2.903, 2.56, and 2.062, respectively. $R_1$ is the "main direction", reflecting the most obvious characteristics of all the 17 antigen factors, which maximizes the information contained in the inducing factors of the production accident.

**Table 7.** Safety production index system of chemical enterprises based on immune model.

| Criterion Layer | Element Layer | Index Layer | | Weight of Criteria Layer |
|---|---|---|---|---|
| Antigen G | Unsafe behavior G1 0.25736 | Staffing G11 | 0.1546 | 0.03977 |
| | | Illegal construction G12 | 0.3364 | 0.08657 |
| | | Personal protection G13 | 0.2364 | 0.06083 |
| | | Personal quality G14 | 0.2727 | 0.07019 |
| | Unsafe state G2 0.24308 | Object failure G21 | 0.1812 | 0.04404 |
| | | Disorderly traffic G22 | 0.1957 | 0.04756 |
| | | Equipment mechanical damage G23 | 0.1884 | 0.04580 |
| | | Violation of distance and placement G24 | 0.2101 | 0.05108 |
| | | Lack of protection G25 | 0.2246 | 0.05461 |
| | Adverse operating environment G3 0.26239 | Major hazard monitoring G31 | 0.3044 | 0.07986 |
| | | Risk warning value exceeds G32 | 0.2957 | 0.07758 |
| | | Environmental pollution G33 | 0.2174 | 0.05704 |
| | | Occupational health hazards G34 | 0.1826 | 0.04792 |
| | Lack of Safety Management G4 0.23717 | Safety education G41 | 0.2315 | 0.05490 |
| | | Security screening governance G42 | 0.2685 | 0.06368 |
| | | Work safety supervision G43 | 0.2593 | 0.06149 |
| | | Safety identification and Emergency G44 | 0.2407 | 0.05710 |
| Antibody B | Safe behavior B1 0.24432 | Personnel on-job management B11 | 0.2000 | 0.04886 |
| | | Behavior safety observation B12 | 0.2091 | 0.05108 |
| | | Personnel safety training B13 | 0.2636 | 0.06441 |
| | | Risk control of hazardous operations B14 | 0.3273 | 0.07996 |
| | Safe state B2 0.28723 | Monitoring and early warning of major hazard sources B21 | 0.3023 | 0.08684 |
| | | Monitoring and warning of high-risk processes B22 | 0.2713 | 0.07793 |
| | | Video surveillance of vital parts B23 | 0.2636 | 0.07570 |
| | | Inherent facility risk management B24 | 0.1628 | 0.04676 |
| | Good working environment B3 0.23232 | Risk monitoring map B31 | 0.2821 | 0.06553 |
| | | Monitoring and warning of toxics B32 | 0.4359 | 0.10127 |
| | | Occupational disease risk monitoring B33 | 0.2821 | 0.06553 |
| | Safety management B4 0.23613 | Risk prevention and control objective management B41 | 0.1691 | 0.03994 |
| | | Risk identification and control B42 | 0.2279 | 0.05382 |
| | | Risk investigation and management B43 | 0.2353 | 0.05556 |
| | | Emergency management B44 | 0.1838 | 0.04341 |
| | | Training learning assessment B45 | 0.1838 | 0.04341 |

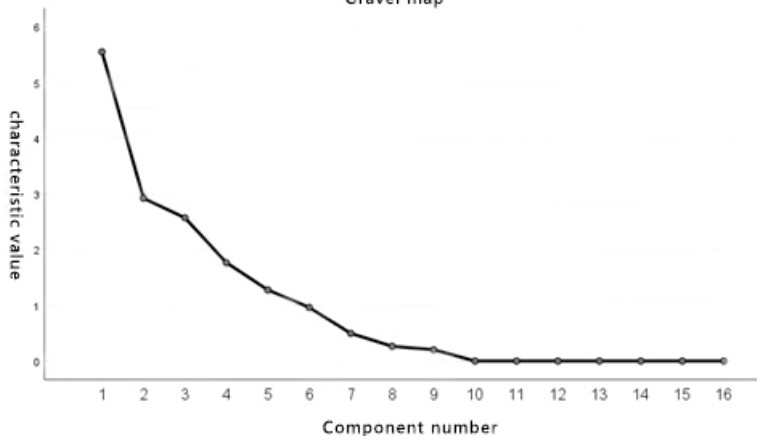

**Figure 4.** Scree plot of extraction of antibody factor components.

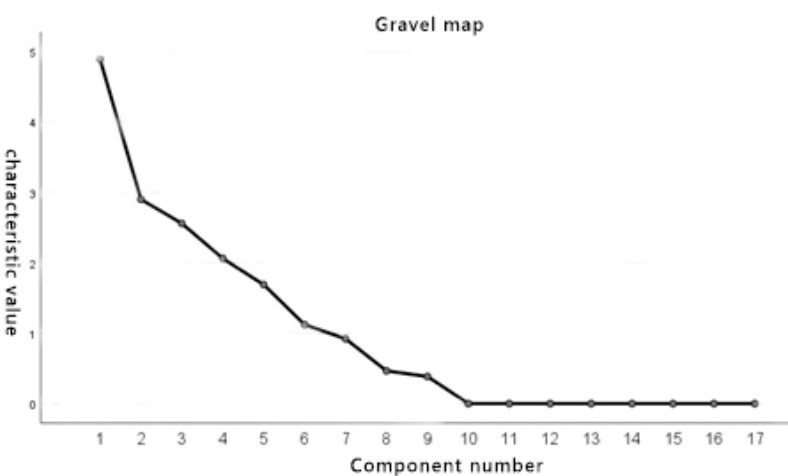

**Figure 5.** Scree plot of extraction of antigenic factor components.

The comprehensive score values of antigen index layer G1–G4, namely the contribution degree $\theta$ of induction effect of safety production accidents, are 2.0656, 2.6283, 2.2955, and 2.3022, and the final score of accident risk maturity of chemical enterprise is 0.5653.

b.    Analysis and evaluation of the safety status of immune system model

According to the antigen–antibody comprehensive score, the early warning and evaluation criteria are established. The definition standard is divided into five grades, [0.0, 0.2], [0.2, 0.4], [0.4, 0.6], [0.6, 0.8], and [0.8, 1.0], corresponding to very low, low, average, high, and very high warning zones, respectively. The higher G value is (closer to 1) and the lower B value is (closer to 0), the higher [G-B] value is, indicating that the safety state of chemical enterprises is worse, and it is more likely to cause production safety accident. On the contrary, lower G value (closer to 0) and higher B value (closer to 1) will lead to higher [B-G] value. This result shows that the better the safety production situation of chemical enterprises, the lower the probability of accident, and the higher the layer of effective immunity.

It can be seen from the left figure of Figure 6 that most of the index values of antigen G layer in the safety immune system of chemical enterprises are >0.5, indicating that the concentration values of various antigens evaluated by experts are generally high, and the influence of antigen indexes extracted from chemical enterprises is strong. The values of G11 and G21 are <0.3, indicating that the staffing and production traffic order are in a safe and stable state. In the antibody index layer, B11 < 0.4, indicating experts agree that companies should ensure the scores of most antibodies at a high layer with strong resistance. According to the figure on the right, the average value of antigen G layer is 2.376, which is at a stable layer, and the influence ranking of each antigen element is G2 > G3 > G4 > G1. The value of antibody B layer fluctuates greatly, and the rank of contribution of each antibody element is B4 > B2> B1> B3.

*5.3. Evaluation Model Construction of Immune Response Ability for Safe Production*

The evaluation index sample matrix A of the evaluation model of immune response ability for chemical enterprises is as follows. Taking the evaluation index "Personnel on duty management/B11" as an example, the sample matrix is: $A_{11}$ = [6,8,6,8,7,6,6,9,5,7].

$$A = \begin{bmatrix} 6 & 3 & 5 & 4 & 4 & 5 & 9 & 9 & 9 & 4 & 9 & 6 & 9 & 6 & 6 & 8 \\ 8 & 1 & 6 & 3 & 5 & 6 & 10 & 7 & 10 & 4 & 10 & 7 & 9 & 5 & 7 & 7 \\ 6 & 1 & 6 & 7 & 3 & 6 & 10 & 8 & 9 & 5 & 10 & 7 & 8 & 5 & 7 & 7 \\ 8 & 2 & 5 & 5 & 5 & 4 & 9 & 8 & 9 & 4 & 10 & 6 & 8 & 6 & 5 & 6 \\ 7 & 4 & 7 & 5 & 6 & 4 & 10 & 7 & 8 & 4 & 9 & 7 & 8 & 7 & 6 & 7 \\ 6 & 3 & 6 & 6 & 4 & 5 & 9 & 9 & 8 & 3 & 10 & 7 & 9 & 5 & 5 & 7 \\ 6 & 2 & 6 & 6 & 4 & 5 & 9 & 8 & 9 & 4 & 9 & 7 & 7 & 4 & 6 & 8 \\ 9 & 2 & 5 & 5 & 5 & 6 & 10 & 8 & 8 & 3 & 10 & 6 & 8 & 6 & 7 & 6 \\ 5 & 1 & 7 & 5 & 5 & 4 & 8 & 7 & 8 & 3 & 10 & 7 & 9 & 7 & 6 & 6 \\ 7 & 2 & 6 & 4 & 3 & 5 & 9 & 8 & 8 & 5 & 9 & 6 & 8 & 5 & 6 & 7 \end{bmatrix}^T$$

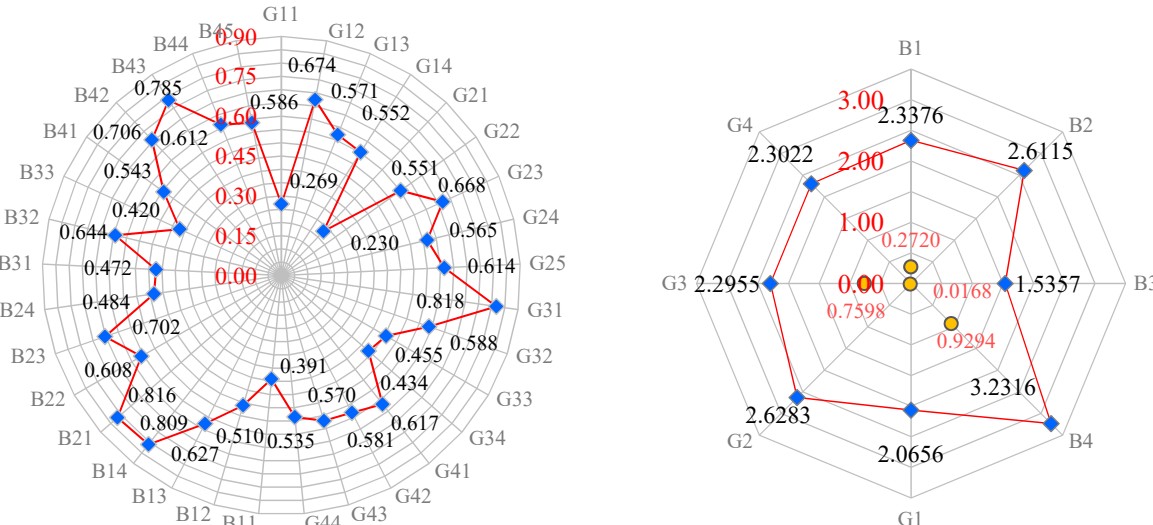

**Figure 6.** Scores of index layer and factor layer of immune system of chemical enterprise in G-B model.

According to Table 2, the grey evaluation coefficients $C_1^e$ of the evaluation index B11 under $e = 1, 2, 3, 4$, and 5 can be obtained: $C_1^{e1} = 6$, $C_1^{e2} = 7.33$, $C_1^{e3} = 7.5$, $C_1^{e1} = 3$, and $C_1^{e5} = 0$, respectively. Then, the grey evaluation weight vector is $Q_1^e = [0.252, 0.308, 0.315, 0.125, 0]$, and the immune response ability assessment result is $R_1^e = 6.37$. Similarly, the evaluation values of B12, B13, and B14 are 3.14, 5.87, and 5.47, respectively. The evaluation result of the element layer $R_{B1} = 5.27$. Similarly, the evaluation results of immune response ability of all indexes at each layer are calculated and listed in Table 8.

It can be concluded that the evaluation result R of immune response ability of chemical enterprise production safety is 6.11, which is judged as "good". It is suggested that the immune system response ability is good on the whole, but the immune response ability corresponding to the unsafe behavior layer/B1 is weak compared with other elements. In the index layer, behavioral safety observation/B12, toxic substance concentration detection and warning/B32, major hazard monitoring, and warning/B21 are the three lowest evaluation categories.

**Table 8.** Evaluation results of evaluation indexes at each layer.

|  | Evaluation Indicator | Grade of the Evaluation | Results |
|---|---|---|---|
| **Index layer** | Personnel on-job management B11 | 6.37 | good |
|  | Behavior safety observation B12 | 3.14 | **poor** |
|  | Personnel safety training B13 | 5.87 | ordinary |
|  | Risk control of hazardous operations B14 | 5.47 | ordinary |
|  | Monitoring and early warning of major hazard sources B21 | 5.14 | **ordinary** |
|  | Monitoring and warning of high-risk processes B22 | 5.28 | ordinary |
|  | Video surveillance of vital parts B23 | 8.03 | very good |
|  | Inherent facility risk management B24 | 7.06 | good |
|  | Risk monitoring map B31 | 7.51 | good |
|  | Monitoring and warning of toxics B32 | 4.84 | **ordinary** |
|  | Occupational disease risk monitoring B33 | 8.36 | very good |
|  | Risk prevention and control objective management B41 | 6.23 | good |
|  | Risk identification and control B42 | 7.31 | good |
|  | Risk investigation and management B43 | 5.73 | general |
|  | Emergency management B44 | 5.97 | general |
|  | Training learning assessment B45 | 6.43 | good |
| **Element factor** | Unsafe behavior of people /B1 | 5.27 | general |
|  | Unsafe state of things /B2 | 6.25 | good |
|  | Poor operating environment /B3 | 6.59 | good |
|  | Safety Management /B4 | 6.34 | good |

## 6. Construction of a Chemical Enterprise Management Platform Based on the Immune Model

Since B12, B32, and B21 are the three lowest indexes, the safety-management platform of chemical enterprises should take appropriate measures to focus on monitoring of human behavior, poison concentration, and major hazard sources. A management platform including personnel positioning, video surveillance, hazard monitoring, and other technologies has been built.

### 6.1. People-Management Platform Based on Location Service

a.    Personnel Location System (PLS)

A personnel-management platform was established based on personnel positioning technology. Considering the actual situation and positioning accuracy of the chemical plant, UWB technology is used to establish personnel positioning system. UWB personnel positioning system is mainly composed of base station with known location, positioning tag, and upper computer (Figure 7). One of the base stations is set as the main base station, and the location data is transmitted to the top computer in real time through serial connection, and the later data processing is carried out through corresponding program software.

b.    Location estimation

The TDOA method is used to calculate the position coordinate to achieve wireless location ranging. As shown in Figure 8, the time difference from the positioning tag to the two nearby positioning base stations is measured first, and then the distance difference can be obtained by multiplying the speed of light. Only two distance differences are required to obtain a set of hyperbolic equations. TDOA method only needs to obtain the time difference, so it does not require the base station synchronize with the tag, which greatly improves the measurement accuracy of the positioning system. Assume that the coordinates of three base stations are 1# $(x_1, y_1)$, 2# $(x_2, y_2)$, and 3# $(x_3, y_3)$, respectively, and tag coordinates are $(x, y)$. The time difference of tag to base stations 1# and 3# is $T = T_3 - T_1$, and then the distance difference between base station 1# and 3# is $D = C \times T$. Consequently, a set of hyperbolic

equations with 1# and 3# as focal points can be obtained. However, this equation contains two unknowns and requires at least three nonlinear equations to compute the position of the target node. Similarly, two other groups of hyperbolic equations with 1# and 2# as focal points and 2# and 3# as focal points can be obtained. Finally, the equation set 7 can be solved to obtain the position of the positioning tag.

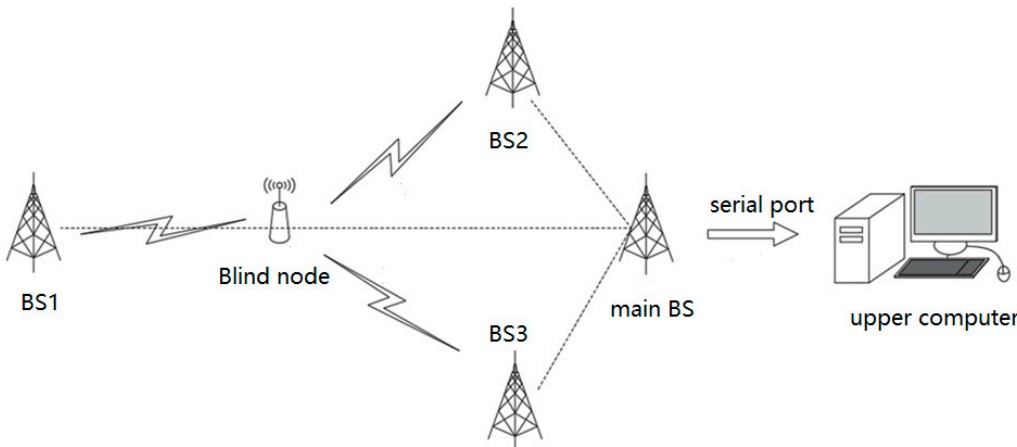

**Figure 7.** Components of a positioning system based on UWB signals.

$$
\begin{cases}
r_2 - r_1 = \sqrt{(x_2 - x)^2 - (y_2 - y)^2} - \sqrt{(x_1 - x)^2 + (y_1 - y)^2} \\
r_3 - r_1 = \sqrt{(x_3 - x)^2 - (y_3 - y)^2} - \sqrt{(x_1 - x)^2 + (y_1 - y)^2} \\
r_3 - r_2 = \sqrt{(x_3 - x)^2 - (y_3 - y)^2} - \sqrt{(x_2 - x)^2 + (y_2 - y)^2}
\end{cases}
\tag{7}
$$

where, $r$ is the search radius. In order to reduce the time error between transmitter and receiver, as well as the influence of clock drift on the calculation accuracy of time difference, the symmetric double-sided bidirectional ranging method is adopted to obtain the signal time of flight (Figure 9). When the positioning system is turned on, the positioning tag sends the matching signal to the nearby base station in real time. If a base station is idle, it will send an initialization signal to the tag, and then the positioning tag is paired with the base station. The tag will then send a signal to the base station containing the address of the base station and record the time $T_1$. When the base station receives the signal with its own address, it will record the receiving time $T_2$ immediately, and after a certain time delay, send the response signal to the tag, and record the sending time $T_3$. After receiving the reaction signal, the tag immediately records the receiving time $T_4$. After a certain time delay, the final signal is sent to the base station, and the sending time $T_5$ is recorded. After receiving the final signal, the base station records the receiving signal time $T_6$, and calculates the two time differences, $t_{reply1} = T_3 - T_2$, $t_{round1} = T_6 - T_3$, and sends the packet containing the two time difference to the tag. The tag can also calculate two time differences according to the recorded timestamp, $t_{reply2} = T_5 - T_4$, $t_{round1} = T_4 - T_1$, and the propagation time of the signal between the tag and the base station can be calculated according to the two time differences sent by the base station, as shown in Formula (8).

$$
TOF = \frac{T_{round1} T_{round2} - T_{reply1} T_{reply2}}{T_{round1} + T_{round2} + T_{round3} + T_{round4}}
\tag{8}
$$

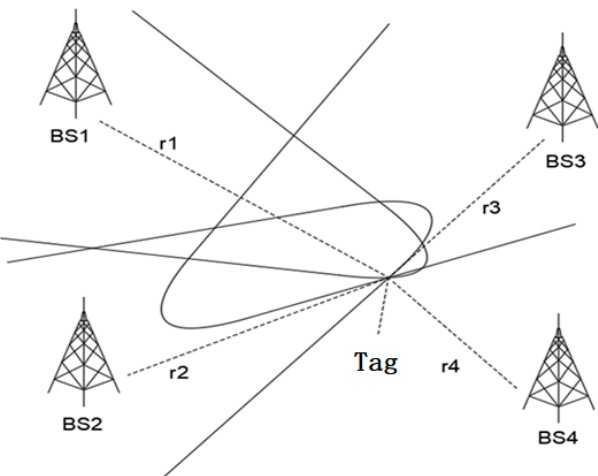

**Figure 8.** TDOA positioning principle.

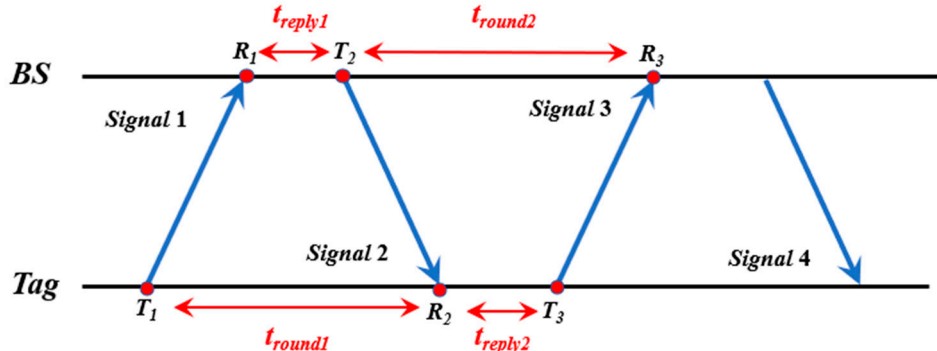

**Figure 9.** Principle of symmetrical bidirectional ranging.

The tag will exchange information with three base stations at the same time. Finally, the packet containing the label address, base station addresses, and signal propagation times will be sent to the upper computer, and the actual location coordinates of the tag will be calculated according to the information. Based on the base station location and signal flight time, the tag position can be solved by establishing a mathematical model. The Chan positioning algorithm is a most widely used positioning solution method based on TDOA, which can calculate high-precision position coordinates with a small amount of calculation. If the coordinates of three stations are $(x_k, y_k)$, then the distance between tag and base station is $R_k$, $k = 1, 2, 3$. The tag coordinate to be obtained is $(x, y)$, then,

$$R_k^2 = (x_k - x_0)^2 + (y_k - y_0)^2 \tag{9}$$

$R_{k,1}$ represents the distance difference between the tag to the $k$ base station and the first base station. It can be obtained accordingly,

$$R_{k,1} = R_k - R_1$$

that is,

$$R_k^2 = (R_{k,1} + R_1)^2 \tag{10}$$

Linearize the nonlinear equations and put Formula 10 into Formula 9 to obtain:

$$R_{k,1}^2 + R_1^2 + 2R_{k,1}R_1 = x_k^2 + x_0^2 - 2x_kx_0 + y_k^2 + y_0^2 - 2y_ky_0 R_1^2 = x_1^2 + x_0^2 - 2x_1x_0 + y_1^2 + y_0^2 - 2y_1y_2$$

By subtracting the two equations, we can obtain:

$$R_{k,1}^2 + 2R_{k,1}R_1 = U_1 - 2x_1x_{k,0} + U_2 - 2y_1y_{k,0}$$

$$U_1 = x_k^2 - x_1^2; x_{k,0} = x_k - x_0$$

$$U_2 = y_k^2 - y_1^2; y_{k,0} = y_k - y_0$$

Taking $x_0$, $y_0$, and $R_1$ as unknown quantities, this becomes a system of linear equations, and the position of the tag will be obtained.

c.    Personnel Management

The establishment of a personnel-management platform based on location service can greatly improve the efficiency of personnel management in the chemical enterprise. It can provide real-time personnel distribution and track activity, and avoid the occurrence of accidents caused by unauthorized intrusion of unrelated personnel in high-risk areas. The platform has the following three functions:

(1)    Personnel information management:

Personnel information management is the basic part of the platform, which is mainly used to store the basic information of staff, including name, employee number, gender, position, authority, and other information.

(2)    Personnel location tracking in real time:

When a staff member enters the work area, their positioning tag will send a UWB signal, and the positioning base station will calculate the position of the tag after receiving the signal, so as to realize the real-time tracking of the staff member's position.

(3)    Historical trace query:

Through the historical track query module, managers can query the movement track of staff and then dynamically understand their activities in the chemical enterprise, and thus strengthen the management of staff.

### 6.2. Visual Monitoring Platform

The visual monitoring platform is mainly used to monitor the real-time changes of various key parameters in the production process and to monitor dangerous operations and high-risk technological processes. It consists of three parts: gas concentration monitoring module, video monitoring module, and equipment operation parameter monitoring module.

a.    Framework design of the platform

The visual monitoring platform mainly completes the four tasks of data collection, data transmission, data display, and data storage, which can realize the timely and accurate transmission of collected data to the upper computer in the monitoring center. According to the actual situation of chemical production, the visual monitoring platform should be able to realize the following functions: real-time transmission of monitoring data, historical data tracing, visual display of monitoring data, and expandable function.

In view of the above demands, the framework of a visual monitoring platform is shown in Figure 10, including two parts, the upper computer and the lower computer. The lower computer is mainly used for parameter monitoring and video picture acquisition, which is divided into gas concentration monitoring system, equipment operation parameter monitoring system, and video monitoring system. Through the data transmission system, the real-time monitoring data will be aggregated to the upper computer. The upper computer is mainly composed of a server and a monitoring PC, which stores and processes the relevant data monitored by various monitoring equipment in real time, and displays them in real time on the screen.

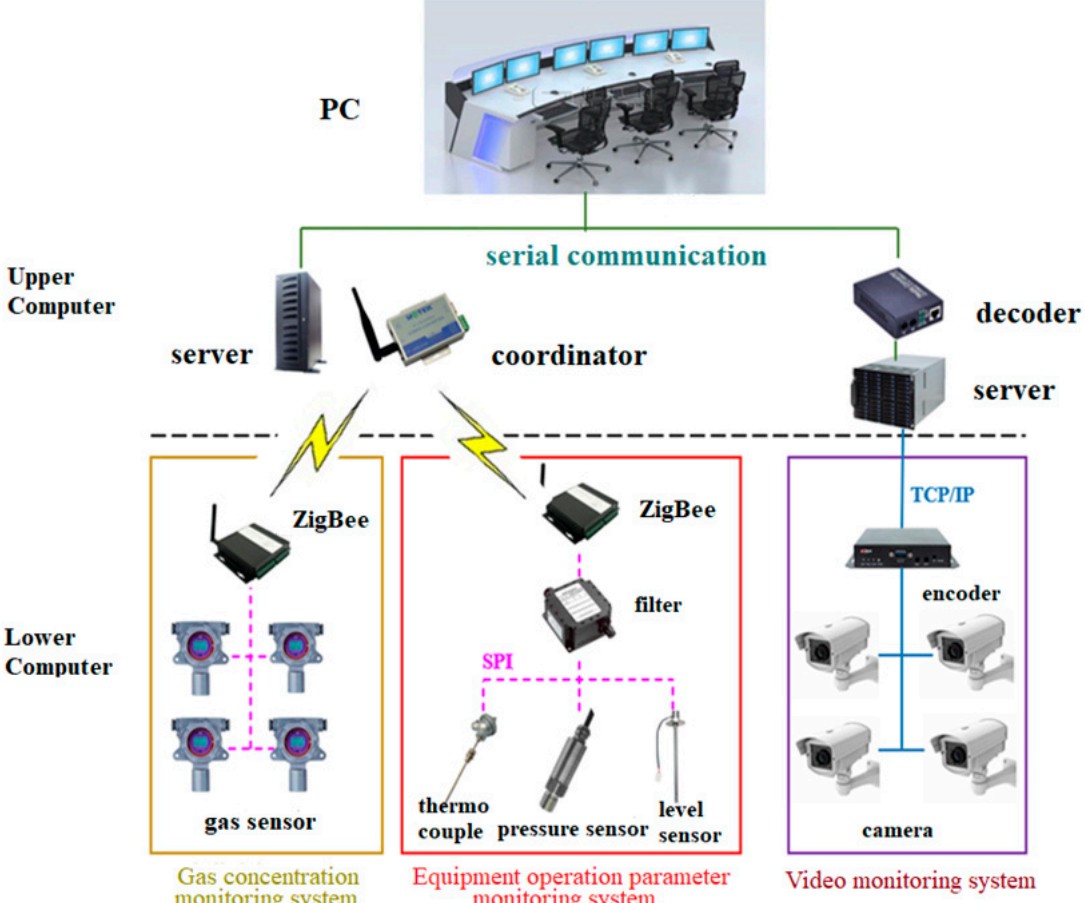

**Figure 10.** General framework of visual monitoring platform.

b.    Gas concentration monitoring system

Based on gas sensors and ZigBee wireless transmission network, a gas concentration monitoring system for real-time monitoring of toxic gas is established, including various gas sensors, a ZigBee module, a microprocessor, and a power supply unit, as shown in Figure 11.

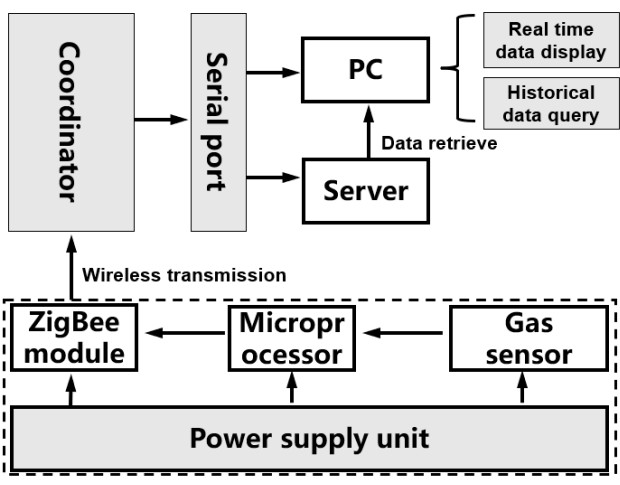

**Gas concentration monitoring system**

**Figure 11.** Structure and function diagram of the gas concentration monitoring system.

The analog signal collected by the gas sensor is converted into digital signal through the microprocessor, and then processed and packaged. On receiving the data packet containing the gas concentration value, the ZigBee module then sends it to the coordinator through wireless transmission. In the gas concentration monitoring system, the coordinator plays the role of constructing ZigBee wireless network, and can transmit the received data packet to the server through serial communication. After receiving the data packet, the server stores and processes the data, and realizes the graphical display of monitoring data on the PC interface.

c.    Equipment operation parameter monitoring system

Temperature, pressure, and liquid layer are the most vital parameters in equipment operation, which need to be monitored in real time to help safety management personnel grasp the real-time changes of relevant parameters. The operation parameter monitoring module consists of sensors, filters, AD transverters, microcontrollers, and a ZigBee module, as shown in Figure 12. Due to the complex working environment of the sensor, the output signal is usually small and easy to be disturbed. Therefore, it is necessary to use the filter to effectively filter and amplify the signal to obtain more accurate monitoring data. Compound filtering method is adopted to process the measured signal, that is, n sampled values are sorted from large to small, with deleting the maximum and minimum values, the arithmetic average value of the remaining n-2 values is obtained, which comprise the final data. The filter then transmits this signal to the AD transverter, which converts the processed analog signal into digital signal, and then the microcontroller encodes the digital signal and packages the data. Parameter monitoring data packets are also sent to the server through ZigBee module, and the server stores and processes the data packets. Finally, the PC software extracts data from the server through serial port and displays various parameter values in real time.

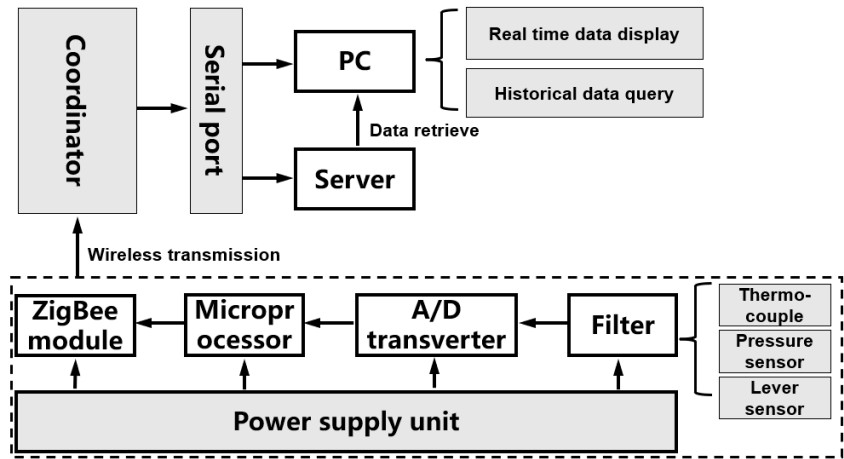

**Figure 12.** Structure and function diagram of the device parameter monitoring system.

d.    Video monitoring system

The video monitoring system of chemical enterprise should have the functions of instant transmission of video signal and data tracking, so as to realize the real-time monitoring of chemical production processing and retrieve historical data in case of emergency. A video monitoring system consists of a camera and a video coder, as shown in Figure 13. The camera first converts the collected video signals into digital signals, and sends them to the video encoder for compression and coding, and then transmits them to the server for storage. After decoder decoding, the PC can display the video picture.

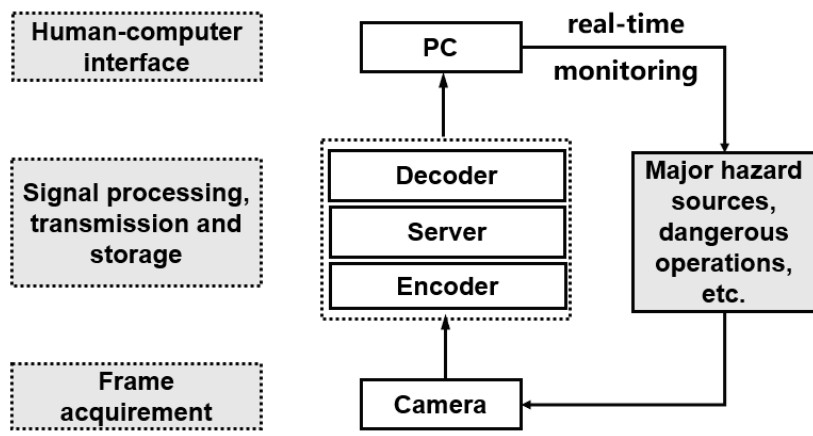

**video monitoring system**

**Figure 13.** Flowchart of the video monitoring system.

*6.3. Danger Warning Platform*

Although a personnel-management platform and a visual monitoring platform have been built, which can track personnel location in real time, monitor toxic gas concentration in the working area, and carry out various operating parameters of equipment, the amount of data is large, and the traditional method of safety management through personnel monitoring has obvious deficiencies in accuracy and efficiency. The danger warning platform can evaluate the current situation in the chemical production process according to the real-time feedback data from the personnel-management platform and the visual monitoring platform, and predict abnormal operations, personnel violations, and maloperation of equipment, in order to prevent accidents at the root.

The establishment of the danger warning platform involves many technical elements, and works in accordance with the sequence of data acquisition, data analysis, and system action. Firstly, it is necessary to identify and classify the "antigen", then select the hazard sources of high-risk degree for real-time monitoring. Then, the real-time monitoring data will be independently analyzed and predicted. If a danger is found, the system will inform the safety management personnel. After receiving the alarm, the safety management personnel will make decisions according to the actual situation, eliminate the hidden danger in time, and achieve the purpose of eliminating the "antigen". According to the demand of chemical plants, the danger warning platform is constructed from two parts: location safety warning module and threshold overlimit warning module.

a.     Location safety warning module

Location safety warning refers to instances when the staff or outsiders enter the dangerous area by mistake or violation in the chemical plant area; the upper computer will send out an alarm message first, and the label worn by the operator will also emit an alarm sound to notify them. The location security warning module is based on UWB personnel positioning; compared with traditional physical isolation, it can realize real-time supervision.

b.     Threshold overlimit warning module

Threshold overrun warning module is the premise of automatic monitoring system. According to the actual situation of chemical production, the safety management personnel set the best parameter range for the monitored gas concentration and various equipment operation parameters. When the monitoring data exceeds the default settings, the risk monitoring platform will immediately start the alarm program.

**7. Conclusions**

(1)     Based on the biological immune system, this paper deeply integrates the immune mechanism and the production safety management of chemical enterprises to form

an accident risk immune correlation principle. According to the actual production situation of chemical enterprises, the evaluation index system of the antigen–antibody model, based on hidden danger and measures in chemical enterprises, has been established, including two systems of antigen G and antibody B with 33 indexes. Antigen G reflects the hidden dangers caused by person, material, environment, and management in the production process of chemical enterprises, while antibody B comprises preventive and control measures for antigen immunity such as education and training, monitoring, and early warning.

(2)    Through the analysis of the antigen–antibody model, the existing problems and unknown hidden dangers in the safety management of the enterprises were excavated, and the immune system model of safety production was established. According to the weight of the improved AHP calculation index system and the contribution degree of each index calculated by the PCA method, the production safety immune evaluation model was established. The results show that the immune maturity of the safe production in chemical enterprises is 0.6114, higher than the accident risk maturity of 0.5653, and the influence ranking of antigen factor G is as follows: G2 > G3 > G4 > G1. The contribution of antibody factor B ranked as B4 > B2 > B1> B3, indicating a clear approach to accident prevention.

(3)    The evaluation model of immune response ability has been designed based on grey system theory to evaluate the effect of the antigen–antibody model of production safety in chemical enterprises. It reveals that production safety has a good immune response ability, but people's unsafe behaviors in the element layer are weak. In its subordinate index layer, only the immune response ability of behavior safety observation/B12 is at a poor layer with the value of 3.14. An evaluation model of immune response ability can effectively distinguish and modify the immune deficiency and vulnerability items of the system, guiding the construction of the informatization management platform for chemical enterprise safety production.

(4)    Targeting the three categories with the lowest evaluation scores in the index layer, combined with the characteristics of chemical production, the personnel-management platform, the visual monitoring platform, and the danger warning platform—based on personnel positioning technology, sensor monitoring technology, video monitoring technology, and internet of things technology—are constructed, which effectively improve the "immune ability" of chemical enterprises, and have practical significance in improving risk monitoring and early warnings in the whole chemical industry.

In this paper, the immune system model of chemical enterprise safety production was applied to the comprehensive assessment of actual chemical safety for human, machine, environment, and management elements. The results show that the model can well determine the hazard tolerance level of chemical enterprises and their development trend. The stability state of the accident immune system is revealed by the maturity level of each accident immune factor. The improved hierarchical analysis method and principal component analysis judgment were used to provide early warning of the risks of chemical production safety. Finally, the direction of continuous improvement and the main ways to prevent and control accidents for the production safety of enterprises were clarified. For example, in Section 6, after identifying the areas of weak hazard tolerance of the enterprise, corresponding protective measures were targeted to enhance the safety immunity level of the enterprise.

The researchers' information on chemical enterprises is limited, meaning that the construction of immunity assessment indicators is not completely comprehensive, and some indicators inevitably have some crossover. There are some indicators in the model that cannot reach a clear quantitative description, which affects the accuracy of the model. In the future, we should consider a more objective and comprehensive way to obtain the parameters of the index system for constructing the assessment capacity and assessment indicators. Meanwhile, the next step is to obtain more objective weights to better reflect the real immune security status of the enterprise. In addition, the current conditions for the

assessment of indicators can only develop relatively general criteria for defining the level. How to set more detailed and specific grade definition criteria and develop a database of countermeasures to optimize antibody items, accurately assess and regulate the level of production safety immunity of chemical enterprises, as well as improve the depth and breadth of research on the similarity between immune mechanisms and safety accidents, are all subjects for further research.

**Author Contributions:** Methodology, X.Y.; formal analysis, B.W.; investigation, X.G.; data curation, Y.W.; resources, O.C.; project administration, J.Z.; writing—review & editing, X.C. All authors have read and agreed to the published version of the manuscript.

**Funding:** This work was supported by the Fundamental Research Funds for the Central University (2019XKQYMS70).

**Institutional Review Board Statement:** Not applicable.

**Informed Consent Statement:** Not applicable.

**Conflicts of Interest:** The authors declare no conflict of interest.

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
