# Peer review of "Construction of Safety-Management Platform for Chemical Enterprises Based on the Immune System Model"

_applsci, doi:10.3390/app12115522_

Round 1
Reviewer 1 Report
This is a paper that has caught my attention. it is certainly interesting, articulated and with stimulating perspectives.
However, I do not hide my difficulties in trying to understand it fully.
In my opinion, it is necessary to supplement the work with a glossary in which the authors clearly describe what they mean when they refer and use medical terms to indicate elements of industrial safety.
Another shortcoming in the paper is a table that encloses all the parameters used with their respective descriptions to avoid redundancy and confusion.
These two models are often cited in the work:” the immune evaluation model of safe production and the immune response ability evaluation model”, the meaning of these titles, in my opinion, represents the same concept: an evaluation of the response system. Is my deduction correct? If so, I would suggest a modification to the model titles based on their real differences, i.e., construction techniques. On the other hand, if I am wrong, then I would ask the authors to insert an explanation that can help in understanding the differences between them.
In paragraph 3, between lines 98 and 102 the same concept is repeated twice.
Also in paragraph 3, in line 104, check the sentence “…, avoid the control of accident infection”. It seems indicates a contradiction.
Here are some first examples of information integration required and concepts or parameters to be described or improved. My suggestion is to review all the work and improve these points to help the interlocutor to be able to linearly follow the logic of the paper, which is definitely worth the attention.
For example, table 1: a clearer description of the 4 levels than that reported between lines 113-118 and detailed of all the various indexes present should be integrated. I recommend using the same terminology to refer to the various levels to avoid confusion, i.e., the first two are called layers, the other two levels. One or the other term should be chosen.
For example, line 127: what does that “each item” refer to? And line 128: the parameters G, m, W what do they refer to?
Or again, in sub-paragraph 4.2.a: a reference is made to “immune factors”. Would these be the level 4 factors of table 1? and the 4 “immune objects” are perhaps antibody B1-B4?
Whereas in eq 1: the Xij parameter is not defined. Please, verify that all parameters are well defined.
etc.
Moreover, please also check the references, I found a repetition of the same work [1] and [21]
Author Response
Dear Editors and Reviewers:
Thank you for your letter and for the reviewers’ comments concerning our manuscript entitled “Construction of Safety Management Platform of Chemical Enterprises based on Immune System Model”. Those comments are all valuable and very helpful for revising and improving our paper, as well as the important guiding significance to our researches. We have studied comments carefully and have made corrections which we hope meet with approval. Revised portion are marked in yellow in the paper. The main corrections in the paper and the responds to the reviewer’s comments are as flowing:
Review 1#:
- In my opinion, it is necessary to supplement the work with a glossary in which the authors clearly describe what they mean when they refer and use medical terms to indicate elements of industrial safety.
Response: The glossary has been provided here, and the explanations of all the parameters has been added in the manuscript.
|
Parameter |
Meaning |
Parameter |
Meaning |
|
G |
antigen |
B |
antibody |
|
Gij |
mean score of the second index |
mij |
scores by experts |
|
Wij |
weight of index layer |
|
mean value of evaluation indexes |
|
Xik/Xij |
value of index |
Sk2 |
standard deviation of indexes |
|
LikT |
a set of unit vectors |
Ri |
immune index after normalization |
|
X |
immune index matrix |
λi |
variance of Ri |
|
βi |
correlation coefficient |
αik |
the ith value in the kth eigenvector |
|
θi |
contribution of the ith index |
Ce |
grey evaluation coefficient |
|
A |
sample matrix |
Qe |
grey evaluation weight vector |
|
e |
grading of evaluation layer |
Re |
grey evaluation result |
|
CR |
random consistency ratio |
CI |
maximum characteristic root |
|
RI |
random consistency |
Rk/rk |
distance between tag and base station |
|
R |
overall immune response ability |
w |
weight matrix |
- These two models are often cited in the work:” the immune evaluation model of safe production and the immune response ability evaluation model”, the meaning of these titles, in my opinion, represents the same concept: an evaluation of the response system. Is my deduction correct? If so, I would suggest a modification to the model titles based on their real differences, i.e., construction techniques. On the other hand, if I am wrong, then I would ask the authors to insert an explanation that can help in understanding the differences between them.
Response: The “immune evaluation model of safe production” and “the immune response ability evaluation model” are not exactly the same model.
There are three models for safe production in chemical enterprises in this paper, namely, immune system model, immune evaluation model, and immune response ability evaluation model.
Immune system model is constructed by analyze the hidden dangers in the existing safety management through antigen-antibody model, including 8 factor layers and 33 index layers; Immune evaluation model is established according to the weight of the improved AHP index system and the contribution degree of each index calculated by principal component analysis (PCA); immune response ability evaluation model is obtained from grey system theory.
The interpretation of these models has been discriminated in the revised abstract.
- In paragraph 3, between lines 98 and 102 the same concept is repeated twice.
Response: Thank you for your correction. We have removed the extra one.
- Also in paragraph 3, in line 104, check the sentence “…, avoid the control of accident infection”. It seems indicates a contradiction.
Response: Thank you for your correction. What I wanted to say is “ avoid from being controlled by the accidental infection”, but we failed to describe it well. We have revised the writing errors and marked them in yellow.
- Table 1: a clearer description of the 4 levels than that reported between lines 113-118 and detailed of all the various indexes present should be integrated. I recommend using the same terminology to refer to the various levels to avoid confusion, i.e., the first two are called layers, the other two levels. One or the other term should be chosen.
Response: Detailed description has been added and marked in yellow. We have changed “levels” into “layers” as well as other confusing expression according to the advice.
- Line 127: what does that “each item” refer to? And line 128: the parameters G, m, W what do they refer to?
Response: "Each" means a score for each index. G is the average index score, m is an expert score, and W is the index layer weight. According to the reviewer’s suggestion, we have described all the parameters in the manuscript to make a better understanding.
- In sub-paragraph 4.2.a: a reference is made to “immune factors”. Would these be the level 4 factors of table 1? and the 4 “immune objects” are perhaps antibody B1-B4?
Response: Yes, the attribution is correct. To prevent misunderstanding, we have changed “immune factors” into “immune indexes” and explain “immune objects” in the manuscript.
- Whereas in eq 1: the Xij parameter is not defined. Please, verify that all parameters are well defined.
Response: We have described all the parameters in the manuscript to make a better understanding.
- Moreover, please also check the references, I found a repetition of the same work [1] and [21]
Response: Thank for your careful review, and the references have been checked.
Reviewer 2 Report
The topic of the article is interesting but its quality should be improved based on the following comments.
- The abstract is poorly written. The abstract should state briefly the purpose of the research, the principal results and major conclusions.
- In the introduction, accident statistics of the chemical industry in China over the past few years should be provided.
- The definition of the immune system is not clear. Some readers may not know such systems well.
- More elaborations on Figure 1 should be provided.
- In Table 1, was the model newly developed in this study? If yes, the theoretical basis should be clearly provided. The development procedure of the model is missing.
- Fore Figure 3, there are five energy levels for the factor. What does each level mean?
- In Table 2, Where do the white functions and their boundaries come from?
- For the expert scoring method, its procedure is not clear. How many experts were included? Who were the experts?
- The theoretical and practical implications, limitations, and future research opportunities were not sufficiently discussed.
- Language issues were found in the manuscript. Proofreading should be conducted.
Author Response
Thank you for your letter and for the reviewers’ comments concerning our manuscript. Those comments are all valuable and very helpful for revising and improving our paper, as well as the important guiding significance to our researches. We have studied comments carefully and have made corrections which we hope meet with approval. Responds are as following:
Review 2#:
- The abstract is poorly written. The abstract should state briefly the purpose of the research, the principal results and major conclusions.
Response: The abstract has been revised according to the indication.
- In the introduction, accident statistics of the chemical industry in China over the past few years should be provided.
Response: The references about accidents of the chemical industry in China over the past few years has been added and marked in yellow.
- The definition of the immune system is not clear. Some readers may not know such systems well.
Response: The definition of immune system is stated in the beginning of Section 1: “Biological immune system is a specific defense system against pathogen invasion, which is a multi-layer adaptive system composed of immune active molecules, immune cells, immune tissues and organs. Once organisms are attacked by pathogens, immune system will quickly produce immune response, identify and kill pathogens, then form immune memory and generate immune feedback. ”
- More elaborations on Figure 1 should be provided.
Response: More illustration for Figure 1 has been added and marked in yellow according to the advice.
- In Table 1, was the model newly developed in this study? If yes, the theoretical basis should be clearly provided. The development procedure of the model is missing.
Response: Thanks for the advice, the development procedure of the model is added and marked in yellow. The development procedure of the models is throughout the whole text, the secondary and explanatory titles can list the procedure. Moreover, the brief methods and processes are provided in the revised abstract as well as the conclusion. By analyzing the feasibility through literature review, Immune system model was firstly constructed by analyze the hidden dangers in the existing safety management through antigen-antibody model, including 8 factor layers and 33 index layers; then immune evaluation model was established according to the weight of the improved AHP index system and the contribution degree of each index calculated by principal component analysis (PCA); Finally, immune response ability evaluation model is obtained from grey system theory, further verify the suitability of immune correlation principle applied in chemical enterprises.
- Fore Figure 3, there are five energy levels for the factor. What does each level mean?
Response: In the manuscript, there’s explanation above Figure 3: “The immune index is divided into 5 layers (Figure 3), namely initial layer (I), definition layer (II), repetition layer (III), continuation layer (â…£) and optimization layer (V).”
- In Table 2, Where do the white functions and their boundaries come from?
Response: Thanks for your comment. Gray system theory determines the whitening weight functions of an object by considering the intrinsic or extrinsic weights among evaluation indexes comprehensively. In 1980s, Deng[1] proposed a gray weighting evaluation method based on the triangular model, and many scholars have continued to improve on this basis. At present, the commonly used whitening weight functions consist of the following four kinds, namely the lower limit measure whitenization weight function, the triangular moderate measure whitenization weight function, the trapezoidal moderate measure whitenization weight function, and the upper bound measure whitenization weight function[2]. Based on the actual needs of this paper, the combination of scoring system and rating system is chosen, and the ten-point system is used to divide the grayscale into [10,8), [8,6), [6,4), [4,2), and [2,0), which correspond to five rating intervals of very good, better, average, poor, and very poor, respectively. In summary, the whitening weight function used in this paper is derived from the typical whitening weight function in the literature and the above fractional interval constitutes the value boundary of the whitening power function.
[1] DENG J L. Grey prediction and decision making. Wuhan:Huazhong University of Science and Technology Press, 1990.(in Chinese)
[2] N. Xie, B. Su, N. Chen, Construction mechanism of whitenization weight function and its application in grey clustering evaluation, Journal of Systems Engineering and Electronics. 30.1 (2019): 121-131.
- For the expert scoring method, its procedure is not clear. How many experts were included? Who were the experts?
Response: We selected experts from enterprises, universities, safety supervision departments and other institutions. On average, the experts have been engaged in work safety for more than 15 years. They gave answers based on their own working background and the actual situation of the enterprise production safety. The score value of the corresponding scoring interval is proportional to the influence degree of the factor layer. The higher the score, the more influence it has on this index. A total of 10 expert questionnaires were issued and 10 valid questionnaires were recalled.
- The theoretical and practical implications, limitations, and future research opportunities were not sufficiently discussed.
Response: We have added the practical implications, limitations, and future research opportunities in the end of the text according to this suggestion.
Round 2
Reviewer 1 Report
Thanks to the authors for following my suggestions and clearing up my initial doubts. There is a lot of work and many different skills behind this paper.
Author Response
Thank you for your review and wish you all the best
Reviewer 2 Report
The authors failed to address my previous comments properly.
- In Table 1, was the model newly developed in this study? If yes, the theoretical basis should be clearly provided. The development procedure of the model is missing.
- It is not clear about the initial layer (I), definition layer (II), repetition layer (III), continuation layer (â…£) and optimization layer (V). No elabortations about them in the manuscript.
- The authors explained the Grey number in Table 2 but failed to explain how the white functions were selected or developed and how their boundaries were defined.
- The expert scoring method details should be added to the revised manuscript. Also, why were only 10 experts involved in the scoring? I do not think 10 experts are enough to generate robust and meaningful results.
- The authors failed to discuss the theoretical and practical implications, limitations, and future research opportunities properly.
Author Response
Dear Reviewer,
Thanks for the careful examination of our paper, and the responses are as follows,
- In Table 1, was the model newly developed in this study? If yes, the theoretical basis should be clearly provided. The development procedure of the model is missing.
Response: Yes, the model is initial one in this study. The model construction steps have been additionally supplemented, and the parameter operation process is detailed in chapters 3.1 and 3.2. Based on the bio-immune mechanism, with the prevention of production safety accidents in chemical enterprises as the goal, the overall analysis of the production structure of the enterprise was firstly conducted from the perspective of bionics. Then, combining the similarity between the biological immune system theory and the safety production system of chemical enterprise, we constructed a hidden danger identification and early warning and risk assessment model for the safety immune system of chemical enterprises. On the basis of the safety information database of chemical enterprises, the results of enterprise hazard source research, chemical production safety regulations and safety expert risk assessment were integrated, and then the antigen/antibody indicators (G11-G44 and B11-B45) of the immune system of production accidents in chemical enterprises were extracted. Finally, hierarchical analysis and expert scoring were applied to determine the weights of indicators set by the evaluation index system to achieve the maturity assessment of production safety immunity of chemical enterprise.
- It is not clear about the initial layer (I), definition layer (II), repetition layer (III), continuation layer (â…£) and optimization layer (V). No elabortations about them in the manuscript.
Response: The abbreviation names and explanations in Figure 3 have been added in detail to the theoretical part of Chapter 3. In order to achieve safe production immune maturity evaluation, a coordinate system is established according to the contribution of immune factors (B) and antigenic factors (G), and the energy level diagram of accident immune-inducing factors is thus drawn in Figure 3. The center of the circle is the core of the immune system, and the outside of the circle is the external environment. According to the contribution of the immune factor to the immune system (the combined antigen-antibody score value), it is divided into five levels: initial level I [0.0,0.2], defined level II [0.2,0.4], repetition level III [0.4,0.6], continuation level IV [0.6,0.8] and optimization level V [0.8,1.0]. For the immunity factor (B), the closest internal level V has the highest energy level, indicating the better safety status of the chemical company. The outermost energy level I is the lowest which is also the highest hazard level. If an accident immune factor is further away from the control target (center of the circle), the lower the degree of control of the antigen by that antibody, i.e., the lower the energy level. For the antigenic factor (G), which induces harm, the energy level diagram is interpreted in the opposite way. Thus, the degree of control of the incident immune system over the harmful factor (antigen G) depends on the positional state of the corresponding immune factor (antibody B), with each antibody being independent, interrelated and mutually constrained.
- The authors explained the Grey number in Table 2 but failed to explain how the white functions were selected or developed and how their boundaries were defined.
Response: At present, the commonly used whitening weight functions consist of the following four kinds, namely the lower limit measure whitenization weight function, the triangular moderate measure whitenization weight function, the trapezoidal moderate measure whitenization weight function, and the upper bound measure whitenization weight function[1]. Combining the above specific meanings of typical whitening weight functions, this paper selects specific whitening weight functions for different ratings according to the actual situation of the research object, very good corresponds to the upper bound measure whitenization weight function, good and ordinary corresponds to the triangular moderate measure whitenization weight function, low corresponds to the trapezoidal moderate measure whitenization weight function, and very low corresponds to the lower limit measure whitenization weight function. For the boundary problem of the function, when the specific form and image of the whitening weight function are determined, the boundary of the function can be simply calculated by the mathematical model.
[1] N. Xie, B. Su, N. Chen, Construction mechanism of whitenization weight function and its application in grey clustering evaluation, Journal of Systems Engineering and Electronics. 30.1 (2019): 121-131.
- The expert scoring method details should be added to the revised manuscript. Also, why were only 10 experts involved in the scoring? I do not think 10 experts are enough to generate robust and meaningful results.
Response: In order to ensure the authority of the score, we sought professional assistance from experts in the expert bank of industry and trade in our province. There are only 200 experts totally, and we only use the scoring from 10 experts who provided timely feedback. Then after we built up models and established formulars, the rest 5 questionnaires were returned. If possible, we will get more experts participate in our project in the future.
- The authors failed to discuss the theoretical and practical implications, limitations, and future research opportunities properly.
Response: The practical implications, limitations, and future research opportunities has been rewritten and added in the end of the manuscript.
Implications
In this paper, the immune system model of chemical enterprise safety production was applied to the comprehensive assessment of actual chemical safety human, machine, environment and management. The results show that the model can well determine the hazard tolerance level of chemical enterprises and their development trend. The stability state of the accident immune system is revealed by the maturity level of each accident immune factor. The improved hierarchical analysis method - principal component analysis judgment was used to provide early warning of the risks of chemical production safety, and finally the direction of continuous improvement and the main ways to prevent and control accidents in the production safety of enterprises are clarified. For example, in Chapter 5, after identifying the areas of weak hazard tolerance of the enterprise, corresponding protective measures are targeted to enhance the safety immunity level of the enterprise.
Limitations and future research
The researcher's information on chemical enterprise is limited, resulting in the construction of immunity assessment indicators is not comprehensive enough, and some indicators inevitably still have some crossover. There are some indicators in the model that cannot reach a clear quantitative description, which affects the accuracy of the model. In the future, we should consider a more objective and comprehensive way to obtain the parameters of the index system for constructing the assessment capacity and assessment indicators. Meanwhile, the next step is to obtain more objective weights to better reflect the real immune security status of the enterprise. In addition, the current conditions for the assessment of indicators can only develop relatively general criteria for defining the level. How to set more detailed and specific grade definition criteria and develop a database of countermeasures to optimize antibody items, accurately assess and regulate the level of production safety immunity of chemical enterprises, as well as improve the depth and breadth of research on the similarity between immune mechanisms and safety accidents, are all subjects to further research.
Round 3
Reviewer 2 Report
The authors addressed my comments properly.